# Mixed-Method Systematic Review and Meta-Analysis of Shared Decision-Making Tools for Cancer Screening

**DOI:** 10.3390/cancers15153867

**Published:** 2023-07-29

**Authors:** Deborah Jael Herrera, Wessel van de Veerdonk, Neamin M. Berhe, Sarah Talboom, Marlon van Loo, Andrea Ruiz Alejos, Allegra Ferrari, Guido Van Hal

**Affiliations:** 1Social Epidemiology and Health Policy (SEHPO), Family Medicine and Population Health (FAMPOP) Department, Faculty of Medicine and Health Sciences, University of Antwerp, Wilrijk, 2610 Antwerp, Belgium; deborah.herrera@student.uantwerpen.be (D.J.H.); wessel.vandeveerdonk@thomasmore.be (W.v.d.V.); neaminmichael@gmail.com (N.M.B.);; 2Expertise Unit People and Wellbeing, Campus Zandpoortvest Thomas More University of Applied Sciences, 2800 Mechelen, Belgium; 3Société Générale de Surveillance (SGS), 2800 Mechelen, Belgium; 4Department of Health Sciences (DISSAL), University of Genoa, Via Pastore 1, 16123 Genoa, Italy

**Keywords:** shared decision-making, cancer screening, vulnerable populations, patient-centered care, decision support techniques, physician–patient relations

## Abstract

**Simple Summary:**

This research was carried out to understand how shared decision-making tools, which facilitate patients and clinicians make decisions based on their values and preferences, can improve decision-making outcomes in cancer screening. The researchers further aimed to explore the preferences of patients and clinicians in terms of the tool’s content, format, and delivery strategies. The review findings showed that SDM tools for cancer screening were more helpful for people facing difficulties in understanding health information or belonging to socially disadvantaged groups, compared to those who have higher educational and socio-economic status and health/language literacy. Moreover, insights from the qualitative synthesis showed that SDM tool preferences for vulnerable populations differ with those of clinicians who are constrained by time during patient consultations. To improve SDM tools, patients and clinicians should collaborate and communicate more. By doing so, they can identify effective delivery strategies that address the needs and preferences of both parties.

**Abstract:**

This review aimed to synthesize evidence on the effectiveness of shared decision-making (SDM) tools for cancer screening and explored the preferences of vulnerable people and clinicians regarding the specific characteristics of the SDM tools. A mixed-method convergent segregated approach was employed, which involved an independent synthesis of quantitative and qualitative data. Articles were systematically selected and screened, resulting in the inclusion and critical appraisal of 55 studies. Results from the meta-analysis revealed that SDM tools were more effective for improving knowledge, reducing decisional conflict, and increasing screening intentions among vulnerable populations compared to non-vulnerable populations. Subgroup analyses showed minimal heterogeneity for decisional conflict outcomes measured over a six-month period. Insights from the qualitative findings revealed the complexities of clinicians’ and vulnerable populations’ preferences for an SDM tool in cancer screening. Vulnerable populations highly preferred SDM tools with relevant information, culturally tailored content, and appropriate communication strategies. Clinicians, on the other hand, highly preferred tools that can be easily integrated into their medical systems for efficient use and can effectively guide their practice for cancer screening while considering patients’ values. Considering the complexities of patients’ and clinicians’ preferences in SDM tool characteristics, fostering collaboration between patients and clinicians during the creation of an SDM tool for cancer screening is essential. This collaboration may ensure effective communication about the specific tool characteristics that best support the needs and preferences of both parties.

## 1. Introduction

Screening is widely considered to be crucial for an improved prevention and prognosis of cancer, as it allows for the early detection of cancer and precursor lesions [1]. In several countries, significant reductions in mortality rates of various cancers, including cervical [2,3], breast [4], lung [5], and colorectal [6,7] have been reported after screening implementation. The availability of screening modalities that provide both primary and secondary prevention through the detection of pre-cancerous lesions has also contributed to the decrease in cancer incidence and disability-adjusted life years [1,8].

Despite the benefits of cancer screenings, an increasing recognition of their limitations and potential for harm has been reported. These limitations include a false sense of safety or anxiety due to test result inaccuracies, the invasiveness of some screening techniques, and the possibility of cancer overdiagnosis and overtreatment [1,9,10]. The possibility that healthy or asymptomatic individuals who undergo screening may be subjected to more harm and complications than benefits from that screening has raised concerns about the individual’s rights to an informed decision [1,11].

To address these concerns, shared decision-making (SDM) tools have emerged as a promising approach to support informed decision-making among people considering cancer screening [12]. These tools are designed to facilitate and support healthcare providers and their patients at different stages of the decision-making process [13]. More precisely, the main purpose of these tools is to assist patients in making healthcare decisions that are well-informed and consistent with their personal values and preferences [14]. SDM tools may take different forms, such as patient decision aids, question prompt lists, interactive websites, which include health information, decision support tools, and personalized guidance to support medical decision-making [15]. Patients must possess adequate levels of health literacy and sufficient knowledge to understand the essential trade-offs between available options and make informed choices to initiate effective SDM [16,17,18]. Thus, SDM tools generally benefit non-vulnerable populations, including those who have sufficient education, a higher health literacy or an ability to actively seek information, and the capability to advocate for their needs [17,19,20,21].

Vulnerable populations, on the other hand, experience lower rates of screening uptake despite being eligible, compared to non-vulnerable populations, due to various barriers, such as language, cost, structural constraints, and cultural factors [22,23,24]. Vulnerable populations in healthcare include those facing income and health disparities, such as individuals with low socio-economic status, limited language and health literacy, immigrants, underinsured individuals, and racial or ethnic minorities [22,23]. Even among those who access screening, their involvement in shared decision-making is often passive, mainly due to limited health literacy and knowledge about cancer and cancer screening [16,17,21].

A recent systematic review and meta-analysis by Yen and colleagues (2021) assessed the SDM interventions’ effectiveness on reducing health inequalities among vulnerable populations [25]. Despite the positive findings regarding patient-reported outcomes for individuals exposed to SDM interventions in the review, several limitations were identified. These limitations included insufficient studies with long-term follow-up periods and a lack of evidence on which specific tool characteristics best support clinicians and vulnerable populations for enhancing the SDM process. Additionally, Stacey and colleagues conducted a systematic review, which demonstrated that SDM tools for cancer screening had positive effects on the SDM process but recommended the need for more evidence on its effect among low health literacy populations [15,26]. Thus, uncertainties remain regarding the discrepancies between effective SDM tools for cancer screening among vulnerable and non-vulnerable individuals, particularly in terms of its effect on SDM and informed decision outcomes.

In light of these knowledge gaps, the primary objective of this review was to synthesize evidence on the effectiveness of SDM tools for cancer screening on shared decision-making and informed decision outcomes in both vulnerable and non-vulnerable people. Furthermore, the review’s secondary objective aimed to explore the preferences of vulnerable populations and clinicians regarding the specific characteristics of the SDM tool. This review provides valuable insights into the differential effects of SDM tools for cancer screening between vulnerable and non-vulnerable people. Furthermore, evidence on the effects of SDM tools on the long-term behavioral outcomes of patients, as well as the importance of a patient-centered and clinician-friendly approach to tool design and implementation are highlighted. Understanding the preferences of both vulnerable people and clinicians regarding SDM tools might substantially contribute towards enhanced inclusivity and equity of the tool’s design and, at the same time, ensure that these tools fit seamlessly into clinical practice.

## 2. Methods

As part of the ORIENT Project (tOwaRds Informed dEcisions iN colorectal cancer screening in Flanders) (https://www.thomasmore.be/en/node/7563 (accessed on 27 July 2023)), this review aimed to synthesize evidence on the shared decision making (SDM) tools for cancer screening. The protocol of this systematic review was registered in the International Prospective Register of Systematic Reviews (PROSPERO registration number: CRD42022382354, https://www.crd.york.ac.uk/PROSPERO/ (accessed on 23 March 2023)). This review adhered to the PRISMA (Preferred Reporting Items for Systematic Reviews and Meta-Analyses) guidelines for transparent and comprehensive reporting [27] (Appendix A).

### 2.1. Search Strategy

Electronic databases were searched from 1 January 2010 to 30 November 2022, including the Cochrane Library and Evidence-based Medicine Reviews via Ovid, MEDLINE via Ovid, EMBASE, and the Web of Science’s Science Citation Index and Social Science Citation Index (Appendix A). The initial electronic search was supplemented by citation mining from eligible studies and relevant systematic reviews.

### 2.2. Eligibility Criteria

This review employed a mixed-method convergent segregated approach, which involved syntheses of quantitative and qualitative literature. For the quantitative review, studies were included if they met the following inclusion criteria: Study population: Studies involving eligible, asymptomatic participants for cancer screening; legal guardians of vulnerable patients; clinicians, including gastroenterologists, general practitioners, primary care providers, nurses, or professionals that are either licensed, in practice, or training.Intervention: SDM tools that were developed and well defined for cancer screening.Comparator: Usual care, simple decision aids, aids, or attention control materials. For before–after study designs, ‘before exposure to any SDM tool’ was considered a sufficient comparator.Outcome: Knowledge, decisional conflict, self-efficacy, the intention to screen, screening uptake/test ordered, decision regret, readiness to decide, and satisfaction with the SDM tool.Study design: Randomized controlled trials, controlled before–after, or quasi-experimental study designs.Other considerations: Studies published between 2010 and 2022, and those that presented baseline information on the study populations making a distinction between vulnerable and non-vulnerable people. Studies were identified to have included vulnerable people when more than 50% of participants had: (1) educational attainment lower than a college degree, (2) immigrant background, or (3) membership of a racial minority [16].

Studies were excluded based on the following criteria:An attrition rate of more than 40%.Small sample sizes (n < 30).Studies that investigated associations relevant to SDM tools but not effectiveness.Developmental, pilot testing, or feasibility studies.

For the qualitative review, studies with the following inclusion criteria were included:Population: Studies involving vulnerable people or clinicians who had experience or were familiar with SDM related to the general patient population.Intervention: No specific intervention, but studies that captured information relevant to any SDM tool for cancer screening.Outcome: Studies that reported findings relevant to vulnerable people’s or clinicians’ preferences in terms of SDM tool content, format, or delivery strategies.Study design: Qualitative or mixed-method studies that used qualitative analysis as part of the methodology, including but not limited to thematic, content, grounded theory analysis, phenomenology, or discourse analysis.

Studies were excluded if they only presented transformed qualitative findings (e.g., from qualitative codes to quantitative data) or if there was insufficient evidence of whether participants belonged to a vulnerable group or were practising clinicians.

### 2.3. Selection Process

Articles retrieved from the electronic databases were exported directly as EndNote X9 files and imported into Rayyan software for de-duplication and screening (https://www.rayyan.ai (accessed on 1 December 2022)). At least two independent reviewers screened the abstract (DJH, WvdV, ST, AF, MvL) and full-text articles (DJH, NB, AA, WvdV, MvL) using well-defined inclusion and exclusion criteria. All reasons for excluding the studies were presented in Appendix A, and the review process is presented in the PRISMA flowchart (Figure 1).

### 2.4. Data Extraction and Management

Prior to the actual data extraction, two independent reviewers (DJH and NB) developed the data extraction forms. The forms underwent pilot testing, following which all relevant data were extracted in duplicate. The data extraction form encompassed more detailed information about the characteristics of the SDM tool, SDM elements, effectiveness, decision aid name, format, theory employed in development, presented screening options, content description, features, and implementation process [28]. The completion of the data extraction forms involved at least two independent review authors (DJH, NB, DH, KA). For qualitative studies, relevant quotations supporting the study findings were compiled into a word document and imported into NVIVO version 12 software for analysis by the two independent review authors (DJH, NB). Any conflicts that arose during the review process were resolved through consensus. A third author (WvdV, MvL, ST) was consulted for arbitration when consensus was not reached. Clarifications were sought by contacting corresponding authors for ambiguous or incomplete studies.

### 2.5. Risk of Bias Assessment

A risk of bias tool (RoB-2) for randomized controlled trials (RoB-2), developed by the Cochrane Collaboration, was used to appraise the quality of the quantitative studies included [29]. At the same time, the Risk of Bias in Non-randomized Studies of Interventions (ROBINS-I) tool was used to appraise non-randomized effectiveness studies [30]. The risk of bias for each eligible study was assessed by two independent review authors (DH and NB), and judgments were categorized as “low risk” of bias, “high risk” of bias, or “unclear risk” of bias for each study. Judgments were reported as “unclear risk” of bias when details about the methods were insufficient, lacking, or not applicable (e.g., for studies where blinding of participants was not possible). All conflicts were resolved by consensus, and a third author (WvdV) was invited to arbitrate where necessary. Furthermore, if reviewers found insufficient information to assess the risk of bias, (e.g., data from trial registries with limited information on methods used in trials), the protocols of the trial were searched to obtain more detailed information on the objectives, design, methodology, statistical considerations, and procedural aspects of a clinical study [31].

### 2.6. Measures of Intervention Effect

For the primary objective, the data analyses were structured using the statistical methods that were developed by Grimshaw and colleagues and were recently applied by Légaré and colleagues [32,33]. Studies were analyzed using means, standard deviations (SD), mean difference (MD), and a 95% confidence interval (CI) for outcomes with continuous measures, specifically knowledge outcomes. However, to summarize the pooled effect for decisional conflict outcomes, the analysis employed standardized mean differences (SMDs) accompanied by 95% confidence intervals (95% CIs). The SMDs were computed as the quotient of the difference in means of outcome measures between groups and the pooled standard deviations (SD) observed across all participants [34]. To evaluate dichotomous outcomes, particularly the intention to screen, the study presented risk ratios (RR) and 95% confidence intervals (CI) derived from event and participant numbers in intervention and comparison groups. These findings were effectively visualized using forest plots. In instances where the standard deviations (SDs) or standard errors (SEs) for differences were unavailable, alternative reported parameters were utilized and imputed according to the Cochrane guidelines [35].

### 2.7. Meta-Analyses

A meta-analysis was conducted using R software version 4.2.2 [36] using the **meta** [37], **metafor** [38], and **metaviz** [39] packages to analyse and visualize data using forest plots and funnel plots. The variations in study population and length of follow-up led to separate analyses between vulnerable and non-vulnerable groups for knowledge and intention to screen outcomes, and length of follow-up, for the decisional conflict outcomes. A random-effects meta-analysis was used to account for the study-level variability in the outcomes [40,41]. The pooling of estimates for decision conflict was conducted using standard mean differences with 95% confidence intervals, while risk ratios with 95% confidence intervals were employed for the intention to screen outcome measures. For studies with more than two interventions, the most comprehensive or novel SDM tool was chosen for the meta-analysis. Similarly, for studies with only multiple comparators, control groups closest to the usual care were selected.

### 2.8. Subgroup Analysis

To address the primary objective, an exploratory subgroup analysis was conducted to examine the presence and magnitude of potential effect-moderating factors that may modify the intervention effects [35]. More precisely, studies were stratified by study participants (vulnerable and non-vulnerable), and duration of intervention (below 6 months and 6 months and above), type of comparator(s) (usual care, attention control), and type of SDM tool based on the targeted cancer screening (prostate, breast, lung, and colorectal cancer screening) (Appendix A). To further investigate the heterogeneity, between-study variations in the meta-analysis were employed. 

### 2.9. Meta-Aggregation

A meta-aggregation using a modified grounded theory analytical approach was used to develop a conceptual framework of the clinicians’ and vulnerable populations’ preferences regarding SDM tool content, forms, and delivery strategies. Subthemes were generated based on the original quotations and generated codes using an iterative process. In addition, the most cited codes across all qualitative studies were presented as common preferences (Section 3.3.3).

### 2.10. Assessment of Reporting Biases

Funnel plots were used to assess the publication bias using the **metavis** package [39] of the free statistical software version 4.2.2 [36] (Appendix A).

### 2.11. Data Synthesis

In this review, variations in outcome measures and statistical analysis of quantitative studies made it necessary to present a narrative synthesis of findings, highlighting the differences between vulnerable and non-vulnerable people. Descriptive statistics were used to summarize study characteristics, particularly the risk of bias. For the qualitative synthesis, Thomas and Harden’s thematic synthesis method was employed to ensure empirical faithfulness to the original data and to allow the integration of reviewers’ independent interpretations of these data [42]. Moreover, themes and subthemes generated from the reported quotations extracted from original studies were visually presented using a conceptual framework. To assess how much confidence to place in the findings on the effectiveness and harms of SDM tools, the ‘Grading of Recommendations, Assessment, Development, and Evaluation’ (GRADE) approach was used by the review authors [43]. Finally, the appraisal of the quality of evidence for qualitative studies was conducted using the GRADE-CERQual (Confidence in the Evidence from the Review of Qualitative research) approach [44] (Appendix A).

## 3. Results

### 3.1. Literature Search

Table 1 and Appendix A present the summarized characteristics of the included studies. In total, 12,713 articles were retrieved, and resulted in 8628 articles after de-duplication. From these articles, titles and abstracts were screened, and 251 articles underwent full-text screening. A total of 55 articles were reviewed, with 35 articles included for the quantitative part (effectiveness studies) of the review and 21 for the qualitative part (Table 1). The mixed-method study of Halley and colleagues was appraised to address both research questions [45].

### 3.2. Study Characteristics

Of the 21 included qualitative studies, a total of 10 studies encompassed both vulnerable individuals and clinicians [79,80,81,82,83,84,85,86,87,88], while 8 studies exclusively focused on vulnerable individuals [45,89,90,91,92,93,94,95] and 3 studies solely involved clinicians [96,97,98] (Appendix A). 

Several studies were conducted in Europe (*n* = 7), including in Spain [87,98], France [80], Portugal [89], Belgium [83], Germany [84], and Denmark [94], as well as in the United States (*n* = 11) [45,81,82,85,88,90,91,92,93,96,97], Canada [95], and England [79].

For the quantitative part, effectiveness studies were predominantly conducted in the United States (*n* = 29). Only a few studies were conducted in Europe (*n* = 4), including in Spain (*n* = 1), and the Netherlands (*n* = 1), the United Kingdom (*n* = 2). Effectiveness studies were also conducted in some areas of Australia (*n* = 2).

Of the 35 studies, 12 evaluated SDM tools for colorectal cancer screening [45,48,52,54,59,62,68,69,70,73,74,99], 11 for prostate cancer screening [46,47,51,57,59,65,66,72,73,75,78], 8 for lung cancer screening [53,55,56,60,63,64,71,77], and 3 for breast cancer screening [49,50,67]. SDM tools included video-based formats (*n* = 10) [45,52,53,54,58,63,64,72,74,77], of which three were combined with a researcher-led coaching session [72], and an online DA and/or booklet [45,47]. Web-based (*n* = 9) [45,48,50,55,56,62,67,68,75] and paper-based formats (n = 12) were also evaluated, including the use of pamphlets, handouts, booklets, and leaflets [49,57,59,60,64,65,70,73,74,76,78,99]. Some unique formats were also evaluated, such as SDM tools using PowerPoint presentations [51] and a software-based format [66]. The readability of the SDM tools ranged from a 5th to 8th grade reading level, with 12 studies reporting this information [46,48,50,51,54,59,65,71,73,74,75,99]. The delivery strategies varied, with fifteen SDM tools being delivered before consultation, nine during consultation, and two after consultation. However, several studies did not clearly state the delivery strategy. The viewing or reading time of SDM tools varied across studies, ranging from 3.51 min (video-based decision aid) [54] to 120 min (video-based decision aid with coaching session for patients and education session for providers) [72].

The characteristics and delivery strategies of SDM tools were mainly described based on (1) their intent or context, (2) the framework, theory, guidelines, or models they were based on, (3) the format and presentation, (4) content, (5) reading level grade, (6) the length of time the patients/providers needed to use, view, or read the tool, and (7) when the tool was delivered (whether it was delivered before, during, and/or after consultation) (Appendix A). 

### 3.3. Meta-Analyses

#### 3.3.1. Knowledge

A total of 27 out of 35 studies assessed knowledge outcomes. Of these, the pooled effect of SDM tools (*n* studies = 15) that evaluated continuous measures on knowledge outcome is presented below (Figure 2A). The SDM tools were found to be highly effective in increasing cancer and cancer screening related-knowledge for all studies (pooled MD = 13.62; 95% CI: 8.25; 18.95; number of studies (*n*) = 15), and for studies involving vulnerable people (MD = 15.44; 95% CI: 9.40; 21.49), but with considerable heterogeneity (I^2^ = 88%; *p* = <0.01) (Figure 2A,B). However, no significant findings were observed on the pooled effects of SDM tools on knowledge outcomes in non-vulnerable people (MD = 13.74; 95% CI: −10.0; 37.48; I^2^ = 94%; *p* < 0.01) (Figure 2C). In addition, our subgroup analysis demonstrated a positive effect, indicating that the use of SDM tools improves knowledge outcomes in colorectal and prostate cancer. However, considerable heterogeneity remains evident in the results (Appendix A). Nonetheless, the funnel plot for knowledge on cancer and cancer screening as an outcome in studies comparing shared decision-making tools to control group, shows that these studies are at low risk of publication bias (Appendix A).

Of note, two studies included in the meta-analysis revealed non-significant findings. However, these studies had comparators that were also SDM tools, which only varied in format [54,73]. Studies that were excluded from the meta-analyses (*n* = 12) had different methodological approaches (e.g., use of non-validated questionnaires), other effect estimates (e.g., proportions instead of mean difference), or varying outcome definitions and comparators. Nonetheless, these studies reported significant improvements in knowledge outcomes, favoring the intervention group (Table 1). 

#### 3.3.2. Decision Conflict

Thirteen out of twenty-one studies that evaluated the effectiveness of SDM tools on decisional conflict outcomes were included in the meta-analyses (Figure 3A). In addition, we stratified studies with an intervention duration of 6 months or less, and above 6 months, to investigate the effects of SDM tools on the long-term decisional conflict outcome (Figure 3B,C). Pooled estimates for all studies presented a significant effect of SDM tools in reducing decision conflict for the participants (SDM = −0.70; 95%CI: −1.23; −0.19; *n* = 13; I^2^ = 97%; *p* < 0.01). Although the favorable effects of SDM tools showed substantial heterogeneity (Figure 3A,B), studies with follow-up periods of more than six months presented no heterogeneity after subgroup analyses (SDM = −0.19; 95% CI: −0.27; −0.11; *n* = 4; I^2^ = 0%; *p* = 0.81) (Figure 3C).

Eight out of twenty-one studies that assessed decisional conflict were excluded from the meta-analyses because of the use of non-validated questionnaires or different effect estimates [50,51,52,53,55,56,62,65,66,78]. Of these, 3 RCTs and 5 non-randomized studies presented significant results favoring the intervention group. The study by Rubel and colleagues, however, which used a Solomon four-group approach, did not reveal significant results [65] (Table 1).

#### 3.3.3. Intention to Screen

Thirteen out of the twenty SDM tool studies that measured screening intentions were included in the meta-analysis (Figure 4). No significant effects were observed for all included studies reporting screening intentions (pooled risk ratios (RR) = 1.11; 95% CI: 0.95; 1.30; I^2^ = 82%; *p* < 0.01, *n* = 13). However, subgroup analysis revealed that SDM tools, compared with the comparison groups, were significantly effective in increasing screening intentions among vulnerable people, with moderate heterogeneity observed (RR = 1.17; 95% CI: 1.07; 1.29; I^2^ = 55%; *p* = 0.02; *n* = 10) (Figure 4A). Furthermore, the intention to screen for colorectal cancer showed a significant increase among the targeted cancer screening population, with only moderate heterogeneity observed (RR = 1.26; 95% CI: 1.12; 1.43; I^2^ = 55%; *p* = 0.04; *n* = 6) (Appendix A).

For studies that measured screening intentions in non-vulnerable people, estimates presented a pooled RR that favored neither the intervention nor the control group in increasing screening intentions (RR = 1.03; 95% CI: 0.44; 2.43; I^2^ = 91%; *p* < 0.01; *n* = 4). Notably, the study by Miller and colleagues reported findings stratified by low and adequate literacy levels, with different SDM tool interventions according to their literacy levels [99]. These findings were, therefore, split during the subgroup analysis. 

The funnel plot for the ‘intention to screen’ for cancer screening in studies comparing after exposure to an SDM tool with control group or before exposure to SDM tool shows a low risk of publication bias (Appendix A). However, the scarcity of points at the bottom of the funnel plot indicates a lack of smaller studies with larger standard errors and less precision. This could be due to smaller studies with null or negative results being less likely to be published or accessible, showing an incomplete representation of the evidence base. It also suggests that studies with positive or statistically significant results are more likely to be published, which could lead to an overestimation of the SDM tool effect. Some points outside the funnel could also be observed, which fall beyond the expected limits of the funnel and suggests the existence of outliers, with extreme effect sizes.

In terms of study design, 9 of the 13 studies were all RCTs with intervention group(s) being compared either to usual care, standard materials, or attention controls. For studies that were excluded from the meta-analysis (*n* = 7), significant findings were reported in four studies, favoring the intervention group instead of the control group [66,68,74,75]. This evidence was mostly observed in studies that included vulnerable people (*n* = 3) [66,68,74], with two of these studies having an overall low risk of bias. On another note, only two of the seven studies excluded from the meta-analysis did not reveal substantial effects of SDM tools on the intention to screen, favoring neither the intervention nor the control group [67,69] (Table 1). 

### 3.4. Narrative Synthesis

#### 3.4.1. Patient–Clinician Screening Discussions

Eleven out of twenty-one studies reported patient–clinician screening discussions (SDM process). Included studies with overall low risks of bias (*n* = 7) show significant effects on patient–clinician screening discussions in favor of the SDM tool interventions, rather than the control groups [54,57,59,61,66,70,74]. Although two studies presented no significant effects of SDM tools, these studies were found to have overall high risks of bias (that is, with at least one high risk of bias in one domain) (Table 1, and Appendix A).

#### 3.4.2. Decisional Regret

Two studies provided information on decisional regret as an outcome measure [67,71]. One of these studies showed significant results, indicating that the intervention was better than the control group [71]. However, this study had an overall high risk of bias, which may limit the validity of its findings. Moreover, the limited number of studies examining decisional regret outcomes implies that there is currently insufficient evidence to draw clear conclusions about their effectiveness in reducing decisional regret.

#### 3.4.3. Self-Efficacy

Two studies reported self-efficacy outcome measures. One of these studies revealed that the qualitative decision support tool was more effective than the framed one [73]. However, this study, conducted by Sheridan and colleagues (2016), only compared four different types of SDM tools that varied in format. As a result, there is a lack of evidence to determine whether SDM tools for cancer screening are more effective in improving self-efficacy outcomes compared with standard materials or usual care. 

### 3.5. Theoretical Framework Based on the Thematic Analysis

A theoretical framework (Figure 5) was developed to summarize the preferences of vulnerable people and clinicians regarding specific SDM tool characteristics for cancer screening. The framework is based on the findings from the thematic analysis and captured the perspectives and experiences of vulnerable people and clinicians.

Three main themes were identified to capture vulnerable people’s preferences: (1) content contextualization, which emphasizes the need for content that is relatable, community-tailored, and culturally sensitive, (2) format relevance, which includes design effectiveness, inclusivity, and functionality, and (3) communication strategies, which refer to the different ways of interacting with the SDM tool during and outside clinical encounters. Three other themes were identified that described the preferences of clinicians regarding SDM tool characteristics, including (1) content quality, (2) format appropriateness, and (3) content delivery. Furthermore, both vulnerable people’s and clinicians’ preferences were supported depending on the (1) usability of the tool’s content and formats, (2) accessibility of the formats and delivery approaches, and (3) the credibility of both the SDM tools and the providers who are responsible for assisting patients in viewing the tool.

### 3.6. Vulnerable People’s Preferences

*Content contextualization*. Several qualitative studies presented the need for SDM tools with relatable and community-tailored content, which respects a patient’s cultural values and beliefs [79,81,82,85,89,95] (Figure 5). The need for relatable content pertains to the information that may be relevant to their history, physiological characteristics (e.g., breast size), and environmental exposures. Vulnerable people who are considering gender-specific cancer screening value information that can help clear up common misunderstandings about cancer and cancer screening [85,89] (Table 2). Some specific examples of common areas of confusion include myths or misinformation about what causes prostate cancer. Another area of confusion is understanding and interpreting risk percentages, which is a complex and confusing topic for these populations. To help address this confusion and provide more personalized and credible information, vulnerable people expressed the desire for information that is easier to understand and more tailored to their individual needs [79,81,82,85,95]. They also preferred to receive information about the risks and benefits of screening tests, as well as the range of costs for screening [45,93,95,97].

*Format relevance*. Eleven out of twenty-one qualitative studies reported findings on vulnerable people’s format preferences for an SDM tool. The most cited preferences were related to an effective design of an SDM tool, for which practices, such as presentation of clear, simple, and eye-catching visuals, use of images, appropriate colors, and illustrations, were highly desirable [79,80,81,85,89,94,97]. Inclusive formats were also preferred, which could cater for people with diverse and multiple races/ethnicities and low literacy levels [82,95]. Common preferences reported to be highly inclusive to these groups were the use of native language [94], layman’s terms [45,80,86,89,93,95,97], presentation of multiple languages [81], and tailoring formats based on the needs of the target populations.

It is also important to note that there are some individuals who highly valued their privacy, and expressed their deep need for formats that could support or protect confidentiality, particularly those who do not feel comfortable about sharing their private information [95]. Finally, preferences were likely influenced by the format’s functionality. For example, some vulnerable people find website or web-based formats to be highly functional as it is easy and fast to navigate specific information, while others find them less functional due to their unfamiliarity with computers [45,80,85,89,90]. It was also noted that paper-based formats were strongly preferred for high accessibility (e.g., can be taken home), and easy distribution, but may provide limited information and explanations about cancer and cancer screening topics [80,83,90,91,95] (Table 2).

*Communication strategies*. Vulnerable individuals expressed a preference for two types of communication strategies when it comes to cancer screening decisions: (1) clinician-assisted strategies during consultations, and (2) strategies beyond clinical encounters. Clinician-assisted strategies include the use of SDM tools with the aid of a trusted clinician or provider before or during consultations [90,93,95,96], while strategies beyond clinical encounters involve the viewing of SDM tools by the patients at home or outside of the clinic, without the assistance of a clinician [80,81,83,91,93,94,96]. Preferences related to this component are highly dependent on the content and format of the SDM tool. For example, some vulnerable people exposed to SDM tools preferred to view them outside of a clinical encounter [45,80,85,90,94,95], while some preferred to view them with their trusted providers [80,90,95]. Furthermore, the credibility of providers highly influences the vulnerable people’s preferences to view or use the SDM tool [79,81,82,93,94]. In both cases, sufficient engagement with either the provider or the tool itself are identified as preferences for vulnerable people, with due consideration given to the patient’s ability to digest and comprehend all the information provided [79,83,85,90].

### 3.7. Clinicians’ Preferences

The content quality in this framework refers to the trustworthiness of the presented data, as well as the ability to guide clinical practices and elucidate one’s values in screening decisions. Format appropriateness refers to the readability (including effective formatting and use of multiple languages) and the flexibility of the formats for possible integration into medical systems. On the other hand, content delivery mainly focuses on the time-saving benefits when delivering content and the ethical conduct while presenting the content (based on efficiency and ethical service provision in cancer screening). In addition, the importance of sufficient expertise on the use of SDM tools (e.g., training clinicians on how to use new SDM tools) was also recognized to be a vital component of effective content delivery.

*Content quality*. The quality of the content is an important consideration in the development of SDM tools. Out of the 21 studies included in the review, 11 studies reported evidence on clinicians’ preferences regarding content quality (Figure 5, Table 2). Trustworthy data were identified as one of the main preferences of clinicians for presenting cancer risk, benefits and/or risks of screening, and for those that utilized statistics, graphic presentations, and multiple languages [80,81,86]. Several clinicians also reported the need for content that elucidates patients’ values and priorities [80,82,83,95]. One clinician in the study by DuBenske and colleagues reported satisfaction with a tool that enabled them to better understand patient priorities and values, which is crucial for effective SDM [82]. Clinicians preferred patients to not solely rely on their recommendations but make informed decisions based on their own priorities and values. The inclusion of content that could guide and harmonize practice with regards to cancer screening was also preferred, which could improve the consistency and quality of care delivered to patients [80,98] (Table 2). 

*Format appropriateness*. Findings show that clinicians favored the readability of the use of scientific or unfamiliar terms [80,86,88]. The presentation of content in a way that is easily understood by both clinicians and patients was emphasized. Clinicians preferred highly flexible tools that could be integrated in their medical software or system [80,97]. This was because the integration of SDM tools into their systems would make them more accessible and usable, leading to better adoption of the intervention. However, it was also important for the information provided to be patient-specific and concise [97]. This implies that SDM tools should be designed in a way that prioritizes readability and accessibility for both clinicians and patients. 

*Content delivery*. Findings show that clinicians placed great importance on the need for SDM tools that could save time during consultations. As a solution for efficient delivery and high accessibility of the SDM tool, clinicians proposed integration of SDM tools in medical systems [79,80,85,86,97]. Moreover, clinicians stressed the importance of delivering content in an ethically correct manner, including proper solicitation of the patient’s consent to use the tool, and providing patients with the option to use the SDM tool or not [79,80]. Finally, the study also found that clinicians recognized the need for training or practical exercises to help gain sufficient expertise before using new SDM tools [83,96,98]. Specifically, they preferred access to links for exercises, as well as opportunities to further understand their patient’s cancer risk, and practice explaining this information to their patients. 

## 4. Discussion

### 4.1. Summary of the Main Findings

This mixed-method systematic review investigated the effectiveness of SDM tools for cancer screening and explored the preferences of vulnerable people and clinicians in terms of the SDM tools’ content, format, and delivery strategies. The meta-analysis results indicated that the use of SDM tools improved patient knowledge and reduced decisional conflict, particularly in the long-term (intervention duration beyond 6 months). However, substantial heterogeneity remained in reported knowledge outcomes, even after subgroup analysis by population, gender-specific screening, and duration of intervention. This is similar to prior studies where substantial heterogeneity continues to exist for outcomes, such as knowledge and intention to screen [15,25,26]. In addition, the review found that vulnerable people exposed to SDM tools had a higher likelihood of screening intention than those exposed to usual care, attention programs, or standard materials. This finding is in opposition to the recent review by Yen and colleagues, which did not report significant effects on the intention to screen among socially disadvantaged populations [25]. Moreover, the communication strategies used in delivering SDM tools to patients and the lack of standardized instruments to assess outcomes could also contribute to the heterogeneity observed in the review.

Finally, review findings show strong evidence for the effectiveness of SDM tools in improving all outcome measures among vulnerable populations, whereas evidence was insufficient for non-vulnerable populations. The findings suggest that the use of SDM tools can be particularly beneficial for vulnerable populations in making informed decisions about cancer screening, compared to individuals with higher education, language/health literacy, and socio-economic status. Notably, relevant results were reported in a systematic review and meta-analysis conducted by Duran and colleagues. The review demonstrated that SDM interventions had a more pronounced positive impact on disadvantaged groups, such as those with lower literacy levels, in comparison to individuals with higher literacy, education, and socio-economic status [100]. Tailored interventions designed to cater to the specific needs of disadvantaged populations proved to be the most effective in promoting shared decision-making in the context of cancer screening.

The qualitative findings revealed that vulnerable people preferred relatable, community-tailored content that respected their cultural values and beliefs. They preferred inclusive formats that catered to diverse races/ethnicities and low literacy levels. The study also found that format functionality was a highly influential characteristic of the SDM tools. This implies that the content and format of SDM tools should be developed with consideration of the cultural and community-specific needs of the target populations, as well as functional characteristics that enhance the usability of these tools. On the other hand, clinicians’ preferences for SDM tools were influenced by the reliability and credibility of the SDM tool content. Clinicians prefer SDM tools that are evidence-based, patient-centered, and which can support efficient service provision. However, informed choice remained controversial, with some clinicians favoring information on screening risks, and others opposing it. Some clinicians were concerned that presenting information on risks would decrease screening participation or create unrest among patients, while other clinicians have shown it to be highly beneficial among their patients as it helped to clarify values and preferences [80,82,83]. However, our quantitative findings revealed that presenting this information using SDM tools was more likely to increase screening participation, particularly among vulnerable people. Furthermore, vulnerable patients reported preferring to be well-informed as it made them “more confident about their ability and feel in control of their health” [94,95,97]. Finally, clinicians who favored including screening risks and neutral content stressed the importance of carefully presenting information in a way that does not generate fear or confusion about screening, while highlighting the benefits of screening [80,83]. This inclusion of risks and neutral content is important in ensuring that patients have access to balanced and accurate information, which can help them make informed decisions about cancer screening.

In the majority of studies (five out of seven, or 71.4%), the reported use of shared decision-making (SDM) tools for non-vulnerable populations involved longer continuous durations (exceeding 30 min). Particularly, a study conducted by Sheridan and colleagues in 2012 revealed that a 120-minute video-based decision aid (DA) with researcher-led coaching did not lead to a significant improvement in patient knowledge [72]. Conversely, interventions specifically tailored for vulnerable populations were generally shorter, typically less than 30 min in duration [46,48,56,57,61,64,66,67,77,99]. These interventions were also designed with a reading level below the 8th grade. Interestingly, these tailored interventions demonstrated a meaningful impact of SDM tools on enhancing knowledge and increasing the intention to undergo screening among vulnerable individuals. This disparity in the length of viewing time and format of SDM interventions between non-vulnerable and vulnerable populations could be attributed to qualitative findings that suggested that patients preferred an effective design of SDM tools presented in a clear, concise, and simple manner, which could influence the effectiveness of the tool. However, it is important to note that the qualitative findings of this study primarily focused on studies involving predominantly vulnerable populations. Therefore, their applicability to non-vulnerable populations may be limited.

### 4.2. Strengths and Limitations

This review provides a comprehensive and holistic view of the impact of SDM tools on cancer screening among vulnerable populations, using a mixed-method convergent segregated approach. The review includes quantitative findings on the differential effect of SDM tools between vulnerable and non-vulnerable people, and further elaborates the reasons why some tools may be effective in improving informed decisions and SDM among vulnerable people and clinicians by understanding their preferences. Moreover, the synthesis of evidence on the effectiveness of SDM tools included highly pragmatic studies conducted in real-world settings, which increases the generalizability of the findings to diverse settings and practices.

Some limitations were also observed in this review. The review only included studies conducted in high-income countries, limiting the generalizability of the findings to other populations and settings. Moreover, some of the study findings could not be pooled in the meta-analysis due to lack of information about the duration of intervention, and the lack of standardized instruments to assess outcomes, specifically, patient knowledge outcomes. This prevented us from further exploring the reasons for heterogeneity in the review’s findings. The optimal timing and frequency of the use of SDM tools, as well as their impact on other outcomes, such as patient satisfaction and adherence to treatment, remains unclear and requires further investigation.

The qualitative synthesis was also limited to untransformed data (that is, from quotations to codes that can be summarized in frequency), which may have limited the presentation and summarization of qualitative data on vulnerable people’s and clinician’s SDM tool preferences. In addition, the review did not explore controversial issues of informed choice among clinicians, which may impact the effectiveness of SDM tools in clinical practice. Several studies were also not specifically designed to capture the preferred SDM tool content, formats, and delivery strategies. Therefore, identified quotations were selected based on the review’s target outcomes, which limits data availability to support the identified themes in the review findings. Nonetheless, we used an iterative process and constant comparison between included studies to reduce confirmation bias, and validation of the generated themes and subthemes was conducted by all review authors. We also analyzed the original authors’ interpretations to further understand the context of the reported quotations.

### 4.3. Implications for Clinical Practice

It is important to note that the effectiveness of SDM tools may vary depending on the population being studied and the length of follow-up. Healthcare providers should consider using SDM tools more prominently among vulnerable populations considering cancer screening. These tools have been shown to be particularly effective in improving knowledge, increasing intention to screen, and reducing decision conflict outcomes for vulnerable populations facing screening decisions.

To ensure that SDM tools are effective and well-received by both target populations, it is essential to incorporate these preferences into the design and implementation of such tools. This involves developing content that is culturally sensitive and relevant to the vulnerable communities, designing formats that are accessible and functional, and incorporating communication strategies that allow for flexible and tailored interaction with the tool. Furthermore, it is crucial to ensure that the SDM tools and the providers who assist patients in using them are both credible and trustworthy. Thus, healthcare professionals should receive training and education on how to effectively use SDM tools and facilitate shared decision-making conversations. Effective communication and education can empower patients to actively participate in the decision-making process, which could lead to better decision-making outcomes in cancer screening.

### 4.4. Implications for Further Research

The lack of standardized instruments to assess outcomes, and specifically patient knowledge, poses a challenge that reduces the comparability of study findings. The development of standardized instruments for assessing patient knowledge in SDM is necessary to enhance the quality of research and improve the outcomes of SDM interventions. Moreover, existing effectiveness studies of SDM tools for cancer screening have only been conducted in high-income countries, which limits the generalizability of results to other populations and settings. Hence, there is a need for further research to evaluate the effectiveness of SDM tools in different populations and settings as well as to capture the clinicians’ and vulnerable populations’ preferences, especially in low- and middle-income countries. Finally, to optimize the use of SDM tools in clinical practice, future studies should also investigate the optimal timing and frequency of SDM tool utilization that best support medical decision-making for both clinicians and people facing screening decisions. This will provide a better understanding of how SDM tools can be adapted to clinicians’ and vulnerable populations’ needs and preferences.

## 5. Conclusions

This mixed-method systematic review provides strong evidence on the beneficial effectiveness of SDM tools among people considering cancer screening, across various shared decision-making outcomes. Moreover, this review demonstrated that SDM tools have a more beneficial impact among vulnerable populations compared to their non-vulnerable counterparts, highlighting that SDM tools may not be universally suitable for all individuals, including non-vulnerable populations. While vulnerable populations may find significant benefits in using SDM tools, other groups, such as those with higher education and health literacy, may have different preferences and decision-making styles. Some individuals may feel confident in making decisions without relying heavily on SDM tools, while others may prefer more interactive approaches. Moreover, the consideration of clinicians’ time constraints is crucial in the implementation of SDM tools in clinical practice. With healthcare professionals facing increasing workloads and limited consultation time, SDM tools need to be seamlessly integrated into their medical systems, ensuring efficient use during patient consultations. By utilizing SDM tools that guide clinicians in facilitating the decision-making process and present information concisely, with a specific focus on vulnerable populations, effective shared decision-making can be achieved without adding undue burden to clinicians’ schedules.

Finally, involving the perspectives and experiences of vulnerable individuals and clinicians in the development of new SDM tools for cancer screening can amplify their beneficial impact and may encourage long-term implementation. However, given the complexities of patients’ and clinicians’ preferences in SDM tool characteristics, fostering collaboration between patients and clinicians during the creation of an SDM tool for cancer screening is essential. This collaborative approach may ensure effective communication about the specific tool characteristics that best support the needs and preferences of both parties.

## Figures and Tables

**Figure 1 cancers-15-03867-f001:**
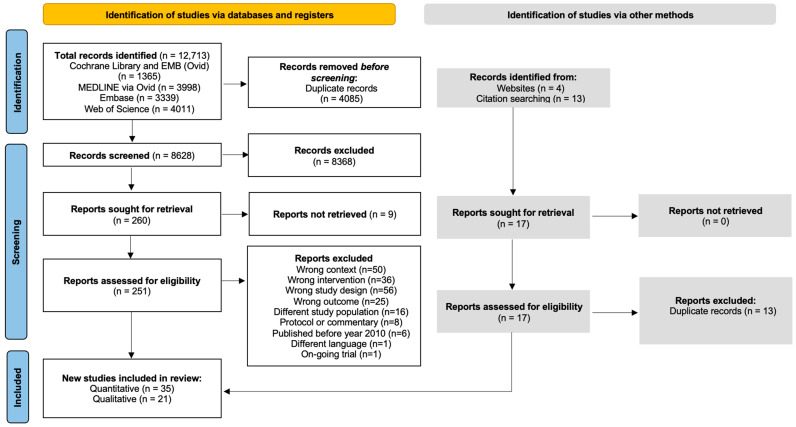
PRISMA flowchart of the study selection process.

**Figure 2 cancers-15-03867-f002:**
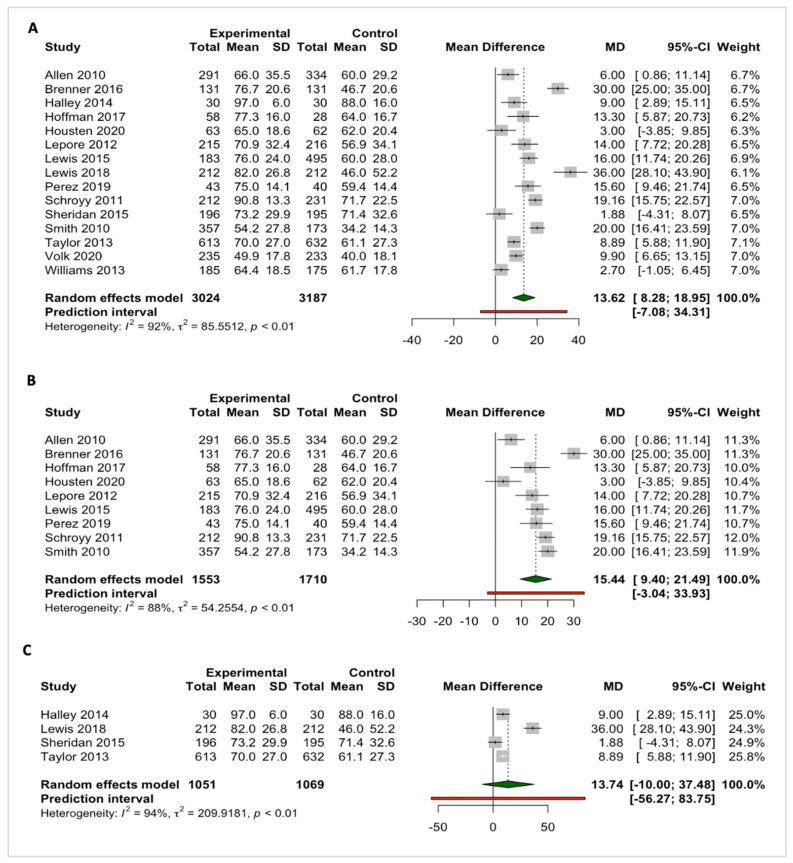
Pooled effects of SDM tools for cancer screening on the ‘knowledge’ outcomes for (**A**) all included participants [45,46,48,52,54,57,58,59,62,68,73,74,75,77,78], (**B**) vul-nerable people [46,48,52,54,57,58,62,68,74], and (**C**) non-vulnerable people [45,59,73,75].

**Figure 3 cancers-15-03867-f003:**
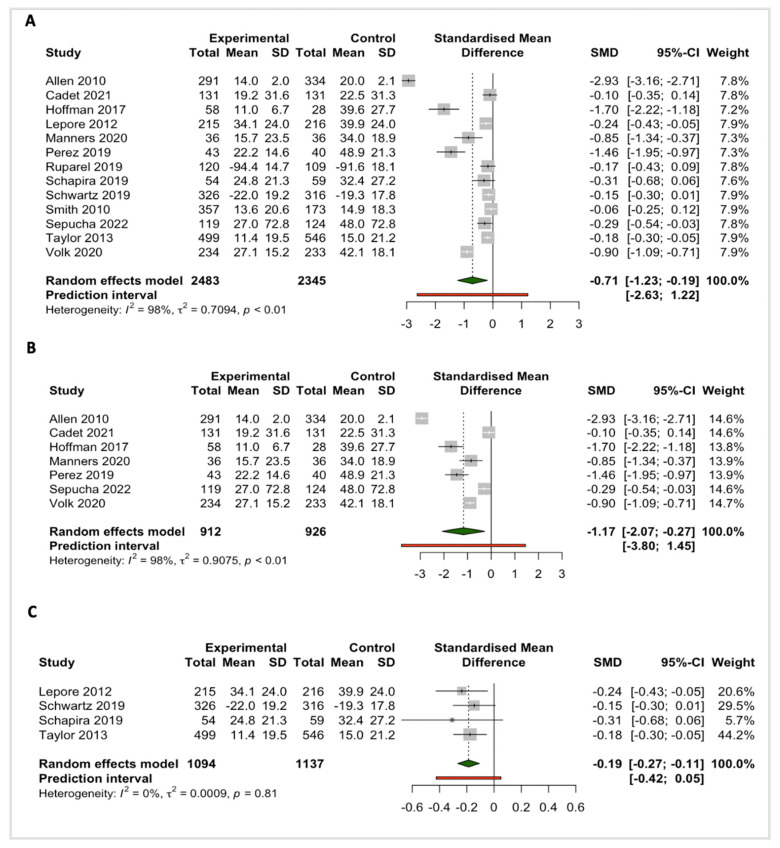
Pooled effects of SDM tools for cancer screening on the ‘decision conflict’ outcomes (**A**) for all included studies [46,52,57,60,62,64,69,70,73,74,75,77,92], (**B**) for studies with an intervention duration at 6 months or less [46,49,52,60,62,70,77], and (**C**) above 6 months (between 8 to 13 months follow-up) [57,69,75,92].

**Figure 4 cancers-15-03867-f004:**
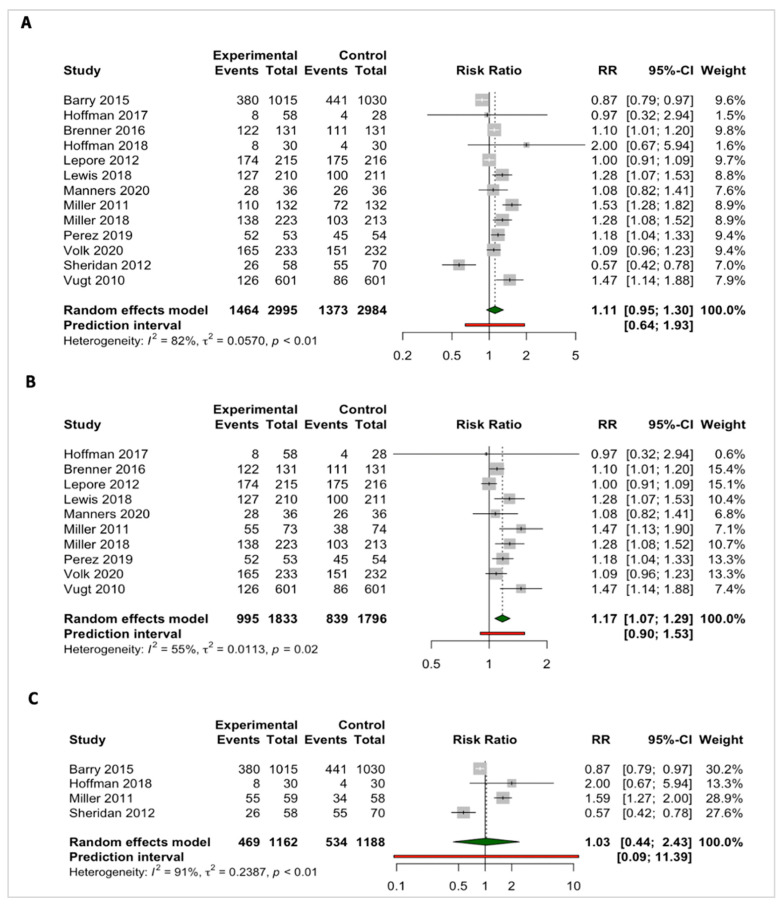
Pooled effects of shared decision-making tools for cancer screening on ‘intention to screen’ outcomes for (**A**) all studies [47,52,57,59,60,61,62,72,76,77,99], (**B**) vulnerable people [48,52,57,59,60,61,62,76,77,99], and (**C**) non-vulnerable people [47,52,72,99].

**Figure 5 cancers-15-03867-f005:**
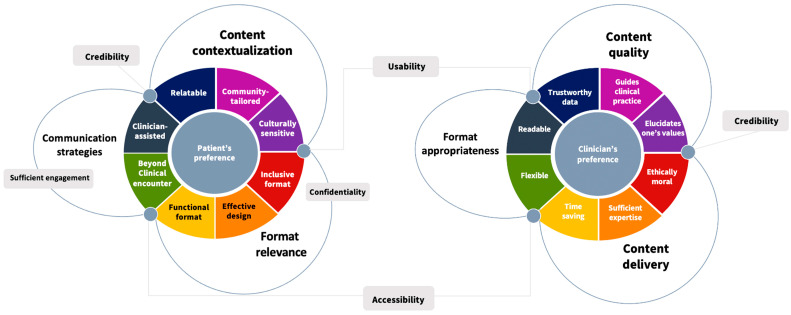
Vulnerable patients and clinicians’ preferences on the specific characteristics of SDM tool. *Outer arches:* These represent the primary overarching themes that broadly capture the perspectives and experiences of vulnerable individuals and clinicians. *Colorful spokes:* These symbolize the subthemes, providing detailed descriptions of the main themes. *Center bore (circle):* This denotes the main outcomes of interest, which are the ‘patient’s preferences’ and ‘clinician’s preferences’ regarding a specific shared decision-making tool. *Outer circles:* These hold the outer arches and signify the important features that influenced the preferences of patients and clinicians concerning the main themes. For example, if main theme 1 is “format relevance” and main theme 2 is “communication strategies,” the circle in between these themes indicates that format relevance and communication strategies should consider the “accessibility” of the tool.

**Table 1 cancers-15-03867-t001:** PICOS Characteristics of included studies that evaluated SDM tools for cancer screening.

Study ID, Country	Study Design	SDM Tool/s Evaluated	Target Group	Comparator	Knowledge	Decision Conflict	Readiness to Decide	Risk Perception	Satisfaction with DA	Screening Uptake/Test Ordered	Intention to Screen	Decisional Regret	SDM Process	Self-Efficacy
Allen 2010 [46], United States	RCT	Computer-tailored decision aid with personalized risk assessment tool	V	Usual care	⇡ ^MA^	⇡ ^MA^	⇡	.	⇡	.	.	.	.	⇡
Barry 2015 [47], UnitedStates	Before–after	“The PSA test: Is it right for you?” DA	NV	NC	.	.	⇡	.	.	.	⇣ ^MA^	.	⇡	.
Brenner 2016 [48], United States	RCT	Spanish (OPCIONES) and English CHOICE	V	Food safety video	⇞ ^MA^	.	.	.	.	⇞	⇞ ^MA^	.	⇞	.
Cadet 2021 [49], United States	Before–after	“Should I continue having mammograms? For women aged 75–84 years,” DA	V	NC	⇡	↮ ^MA^	.	.	⇡	.	.	.	.	.
Eden 2015 [50], United States	Before–after	Mobile application “Mammopad” DA	NV	NC	.	⇡	.	.	.	.	↮	.	.	⇡
Gokce 2017 [51], United States	Before–after	ACS (American Cancer Society) DA	NV	NC	⇡	⇡	.	.	.	.	.	.	.	.
Halley 2015 [45], United States	RCT	Web-based DESI (the decision support intervention)	NV	DVD-first DESI	⇡ ^MA^	.	.	.	.	.	.	.	.	.
Hoffman 2017 [52], United States	RCT	Video-based PtDA	V	Attention control video	↑ ^MA^	↑	.	.	.	↮	↮	.	.	.
Hoffman 2018 [53], United States	Before–after	“Lung Cancer Screening: Is it Right for Me?” Web -based DA	V	NC	⇡	⇡	.	.	.	.	⇡ ^MA^	.	⇡	.
Housten 2020 [54], United States	RCT (3-arm)	Animated video DA and status-video DA	V	Audio–booklet DA	↮ ^MA^	.	.	.	.	.	.	.	.	.
Lau 2015 [55], United States	Before–after	“Lung Cancer Screening: Should I get screened?” web- based DA	NV	NC	⇡	⇡	.	.	⇡	.	⇡ ^MA^	.	.	.
Lau 2021, United States [56]	Before–after	Modified “Lung Cancer Screening: Should I get screened?” web-based DA	V	NC	⇡	⇡	.	.	⇡ **	.	.	.	.	.
Lepore 2012, United States [57]	RCT	“Prostate Cancer: Your Life-You Decide” pamphlet with tailored telephone education	V	Attention control	↑ ^MA^	↑ ^MA^	.	.	.	↮ *	↮ ^MA^	.	↑	.
Lewis 2015, United States [58]	RCT	PSA (prostate-specific antigen)-based DESI + SMA	NV	SMA invitation	⇟ ^MA^	.	.	.	.	.	.	.	.	.
Lewis 2018, United States [59]	RCT	‘‘Making a Decision about Colon Cancer Screening” paper-based DA	NV	Attention control	↑ ^MA^	.	↑	.	.	.	↑ ^MA^	.	↑	.
Manners 2020, Australia [60]	Before–after (quasi)	PtDA + PLCOm2012 risk estimates	V	NC	.	↮ ^MA^	.	.	.	.	↮ ^MA^	.	.	.
Miller 2018 [61], United States	RCT	mPATH-CRC DA	V	Control program	.	.	.	.	.	↑	↑ ^MA^	.	↑	.
Perestelo-Perez 2019 [62], Spain	RCT	Web-based DA	V	Usual care	↑ ^MA^	↑	.	.	.	.	↑ ^MA^	.	.	.
Reuland 2018 [63], United States	Before–after	Video-based DA	NV	NC	⇡	.	.	.	⇡ **	.	.	.	.	.
Ruparel 2019 [64], United Kingdom	RCT	Information film + information booklet DA	V	Booklet alone	⇡	⇡ ^MA^	.	.	⇡	.	.	.	.	.
Rubel 2010 [65], United States	Solomon four- group	CDC-developed PCa screening DA	V	Usual care	↑	↮	.	.	.	.	.	.	.	.
Salkeld 2016 [66], Australia	RCT	Annalisa software-personalized decision support tool	V	Annalisa fixed attribute	.	↑	.	.	.	.	↑	.	↑	.
Schapira 2019 [67], United States	RCT	BCS (breast cancer screening)–PtDA	V	Usual care	↑	↮ ^MA^	.	↮	.	↮ *	↮	↮	.	.
Schroy 2011 [68], United States	RCT (3-arm)	Web-based DA + “Your Disease Risk (YDR)”	V	Generic lifestyle website	⇟ ^MA^	.	.	.	.	⇟	⇟	.	⇟	.
Schwartz 2019 [69], United States	RCT	Quantitative DA	V	Verbal DA	.	↮ ^MA^	.	↑	.	↮ *	↮	.	.	.
Sepucha 2022 [70], United States	RCT	Decision worksheet + telephone session	V	Usual care	.	↑ ^MA^	.	.	.	↑ *	.	.	↑	.
Sferra 2021 [71], United States	RCT	Option grid decision support tool	V	Shouldiscreen.com DA	⇟	.	.	.	.	.	.	⇟	↮	.
Sheridan 2012 [72], United States	RCT	Video-based DA + patient coaching + provider education	NV	Highway safety attention control	⇟	.	.	.	.	⇞	⇞ ^MA^	.	↮	.
Sheridan 2016 [73], United States	RCT (4-arm)	Framed decision support sheet	NV	Qualitative decision support sheet	↮ ^MA^	.	.	↮	.	.	↮	.	.	↓
Smith 2010 [74], Australia	RCT (3-arm)	Booklet + DVD-based DA	V	Standard booklet	↑ ^MA^	↑ ^MA^	.	.	.	↑	↑	.	↑	↮
Taylor 2013 [75], United States	RCT (3-arm)	PCa Web-based DA	NV	Usual care	⇡ ^MA^	⇡ ^MA^	.	.	.	↮	⇡	.	.	.
van Vugt 2010 [76], Netherlands	Before–after	Leaflet PRI (personalized risk indicator)	V	NC	⇡	.	.	.	.	.	⇡ ^MA^	.	⇡	.
Volk 2020 [77], United States	RCT	Video- or DVD-based patient decision aid	V	Standard material	↑ ^MA^	↑ ^MA^	↑	.	.	.	↮ ^MA^	.	↑	.
Williams 2013 [78], United States	RCT (4-arm)	Printed-based DA	NV	Usual care	↑ ^MA^	↑	.	.	.	.	.	.	.	.

**Abbreviations**: DA = decision aid; RCT: randomized controlled trial; V = vulnerable populations; NV = non-vulnerable populations; ML = mixed-literacy population; NC = no comparator; SMA: shared medical appointment; ^MA^ = meta-analyzed. **Symbols**: . = not reported; ↮ = not statistically significant; ↓ = reduced outcome measure with low risk of bias in most domains; ⇣ = reduced outcome measure with unclear to high risk of bias in at least two domains; ↑ = in favor of SDM tool with low risk of bias in most domains; ⇡ = in favor of SDM tool with at unclear or high risk of bias in most domains; ⇞ = in favor of SDM tool with high risk of bias in at least one of the domains; ⇟ = reduced outcome measure with high risk of bias in at least one domain; * = patient–clinician discussions; ** = DA acceptability/preferred the DA/viewed DA as useful.

**Table 2 cancers-15-03867-t002:** Summary of the most common content- and format preferences of vulnerable people and clinicians.

Topic	Studies	Common Preferences (Presented in Codes)
**Vulnerable People**		
Information-specific preferences	*n* = 13 [80,81,83,84,85,87,88,89,90,91,93,94,95]	Key factors related to cancer risk and prevention; Effectiveness and availability of tests for cancer diagnosis and treatment and their risks and benefits; Common confusions and frequently asked questions about cancer and cancer screening; Range of costs of screening; Options of having additional information.
Format-specific preferences	*n* = 13 [45,79,80,81,84,85,89,90,91,93,94,95,97]	Visually clear, simple yet attractive; Use of images, appropriate colors, and illustrations to convey the messages; Presented in simple/native/layman’s language; Presented in multiple languages; A format that is tailored to the needs of the target populations; A tool [format] that is highly functional and has desirable features; Website or web-based format for easy navigation *; DVD or video-based format for sequential presentation of information **; Paper-based formats for high accessibility and easy distribution.
Delivery-specific preferences	*n* = 11 [79,80,81,82,83,84,85,90,91,94,95]	Provision of SDM tools that can be viewed at home; Sufficient time in viewing the tools before consultation or screening discussions; Easily accessible tools before, and/or after consultation; SDM tools to be a part of conversations during consultations; Guidance in viewing the SDM tool with someone they trust (e.g., majority preferred to be guided by nurses rather than doctors).
**Clinicians**		
Content-specific preferences	Studies: (*n* = 8) [79,80,82,83,85,86,97,98]	Use of evidence-based information; Information on risks that does not generate fear or confusion about screening; A clear emphasis on the benefits of screening; Information that prepares patients for discussion with clinicians; Neutral content.
Format-specific preferences	*n* = 8 [79,80,83,85,86,88,97,98]	Use of visuals, such as photos and illustrations presented in multiple languages; Risks presented in a way that is easy to interpret; Formats that speed things up by incorporative tools in clinicians’ electronic systems; Formats with additional features, such as reminders and note taking options; Formats that are highly accessible.
Delivery-specific preferences	*n* = 6 [79,80,83,86,96,97]	SDM tools to be available in the waiting rooms; Use of SDM tools should be a choice with proper consenting of the patients; Clinicians to be trained in using the new SDM tool(s).

* Highly preferred by young to adult populations, ** Highly preferred by older populations and those seeking basic information.

## Data Availability

Data extracted and synthesized are available from the corresponding author upon request.

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
