# Peer review of "Mixed-Method Systematic Review and Meta-Analysis of Shared Decision-Making Tools for Cancer Screening"

_cancers, 2023, doi:10.3390/cancers15153867_

Round 1

Reviewer 1 Report

Interesting systematic review about the effectiveness of shared decision-making tools for cancer screening. Novel approach.

This research was done to understand how shared decision-making tools, which facilitate patients and clinicians make decisions based on their values and preferences, can improve decision-making outcomes in cancer screening. Relevant topic, specially in the field of cancer screening, provides a comprehensive on How share decision-making can be used in cancer screening among vulnerable populations. It would be great if could provide more comprehensive text. The conclusions consistent with the evidence and arguments presented and address the main question posed. No further comments.

Author Response

Response to Reviewer 1 Comments

Thank you for your insightful review of our systematic review on shared decision-making (SDM) tools for cancer screening. We appreciate your positive feedback on our novel approach to this important topic.

Based on your comments, we have made significant improvements to the manuscript. We provided a more comprehensive definition of vulnerable and non-vulnerable populations in the Introduction. We also clarified the conceptual framework in Figure 5 for better understanding.

Regarding results and conclusions, we added findings on knowledge and intention to screen outcomes. We discussed the complexities of preferences among clinicians and vulnerable populations, emphasizing that SDM tools may benefit vulnerable populations more than non-vulnerable ones, while considering clinicians' time constraints. Please find the detailed responses below and the detailed revision in the attached manuscript.

We value your contribution and believe these revisions have strengthened the manuscript. Thank you for your continued support.

Sincerely,

Comments and Suggestions for Authors

Interesting systematic review about the effectiveness of shared decision-making tools for cancer screening. Novel approach.

Point 1: It This research was done to understand how shared decision-making tools, which facilitate patients and clinicians make decisions based on their values and preferences, can improve decision-making outcomes in cancer screening. Relevant topic, especially in the field of cancer screening, provides a comprehensive on how share decision-making can be used in cancer screening among vulnerable populations.

It would be great if could provide more comprehensive text. The conclusions consistent with the evidence and arguments presented and address the main question posed. No further comments.

Response 1: Thank you very much for your generous commendations! Indeed providing a more comprehensive test would enhance the quality of this review. Please see attached manuscript revision, page 2-3, line 43, and line 65-79, for a more elaborate definition of vulnerable and non-vulnerable populations.

Figure 5 of the results section has been improved and clearly presented with the addition of legends to explain the meaning of the conceptual framework.

Moreover, results and conclusions have been improve.

The following results were added:

  • Revised manuscript, page 15, line 345-347 for knowledge outcomes: Firstly, our subgroup analysis demonstrated a positive effect, indicating that the use of SDM tools improves knowledge outcomes in colorectal and prostate cancer screening. However, considerable heterogeneity remains evident in the results.
  • Page 18, line 395-398 for intention to screen, by targeted cancer screening results: Secondly, SDM tools increased the intention to screen outcomes among people considering colorectal cancer screening, with a moderate level of heterogeneity observed.
  • However, no significant effects were observed on intention to screen outcomes for other cancer types.
  • Finally, there is insufficient evidence for other outcomes between different cancer types.

Discussions improved:

  • Please see attached manuscript revision, page 25, line 621-631

Finally, implications were revised to emphasize that SDM tools may have had more benefits in vulnerable populations than in non-vulnerable populations (Please see page 1 line 16-24, and line 30-43 and pages 28-29, line 756-778). We highlighted that these tools might not have been the best fit for everyone, particularly considering clinicians' time constraints.

Sincerely,

Deborah Jael Herrera

Reviewer 2 Report

What vulnerable populations? Define this

Intro –

First sentence – should read improved prevention and prognosis. Oddly worded currently

What is a vulnerable vs non vulnerable population? I think this needs better defined (defined in eligibility criteria but would also greatly help to be defined in the intro)

Figure 5 – not sure that I really understand this figure/its conveying what you want it to? Is this both broad categories and then micro specifics under each category for what is preferred?  This feels confusing to me.

What are the ** in table 2??

In the objective of the paper, you note:

 “the primary objective of this review was to synthesize the evidence on the effectiveness of SDM tools for cancer screening on shared decision making or informed decision outcomes”

-          Isn’t that an ‘and’ ?  if not, can you delineate what falls into each bucket?

Is there any difference between cancer types and outcomes?  i.e. was SDM better for breast and worse in prostate amongst populations?

Would edit implications to state more heavily that it appears that SDM tools may have more benefit in vulnerable than non vulnerable populations – there was no change in intention to screen among non vulnerable populations and in a world where all clinicians are strapped for time, would highlight that actually these tools may not be best for everyone. That feels really important to me.

Author Response

Response to Reviewer 2 Comments

Thank you for your thoughtful and constructive feedback on our systematic review regarding shared decision-making (SDM) tools for cancer screening. We have carefully considered your comments and made significant improvements to the manuscript based on your suggestions.

We have addressed the concern about defining vulnerable populations in both the Introduction and Eligibility Criteria sections. Furthermore, Figure 5 has been revised to include additional legends for better clarity.

Regarding Table 2, we have provided a clear explanation of the ** notation in the legend to help readers understand the specific tool characteristics represented.

In response to your inquiry about the primary objective of the paper, we have clarified that the review aimed to synthesize evidence on the effectiveness of SDM tools for cancer screening in both the shared decision-making process and informed decision outcomes.

Regarding cancer types and outcomes, we have presented detailed subgroup analysis results, particularly for colorectal and prostate cancer screenings. However, insufficient evidence was found for other cancer types.

Lastly, we have revised the implications to emphasize the potential benefits of SDM tools for vulnerable populations and highlight the need to consider clinicians' time constraints when implementing these tools in clinical practice.

We greatly appreciate your valuable input, and we believe these revisions have strengthened the manuscript. Thank you for your time and consideration.

Please find the point-by-point response below and the detailed revision in the attached manuscript.

Sincerely,

Deborah Jael Herrera 

Comments and Suggestions for Authors

Point 1: What vulnerable populations? Define this

  • Intro – First sentence – should read improved prevention and prognosis. Oddly worded currently

Response 1: I have reworded it to "improved prevention and prognosis...". Please refer to page 2, line 43 of the attached manuscript revision to review the changes.

Point 2: What is a vulnerable vs non vulnerable population? I think this needs better defined (defined in eligibility criteria but would also greatly help to be defined in the intro)

Response 2: Thank you for the valuable suggestion. We have now included the description of vulnerable and non-vulnerable populations in the Introduction section. Please refer to page 2, lines 65-79 of the attached manuscript revision to review the updates.

Point 3: Figure 5 – not sure that I really understand this figure/its conveying what you want it to? Is this both broad categories and then micro specifics under each category for what is preferred?  This feels confusing to me.

Response 3: Yes, indeed, it can be confusing. Thank you for this very helpful feedback. The outer categories are the main/broad themes that captured the subthemes, which are presented in the inner wheels of Figure 5. To make this easier to follow, we have added additional legends to guide readers when viewing this figure. Please see the attached manuscript, page 19.

Point 4: What are the ** in table 2??

Response 4: We have incorporated a legend into Table 2 to provide a clear explanation for **. The updated legend in Table 2 now includes the following information:

* Highly preferred by young to adult populations, ** Highly preferred by older populations and those seeking basic information.

Please refer to Table 2 in the attached manuscript revision to view the changes.

Point 5: In the objective of the paper, you note: “the primary objective of this review was to synthesize the evidence on the effectiveness of SDM tools for cancer screening on shared decision making or informed decision outcomes” Isn’t that an ‘and’ ?  if not, can you delineate what falls into each bucket?

Response 5: You are correct; the sentence does imply an 'and' between "shared decision making" and "informed decision outcomes." Here's the delineation of what falls into each bucket:

Shared Decision Making (SDM): This refers to the process in which patients and healthcare professionals work together to make decisions about treatment options, taking into account the patient's preferences, values, and the best available evidence.

Informed Decision Outcomes: This encompasses the results or consequences of shared decision-making. It includes the outcomes related to the patient's understanding of the information presented (e.g., knowledge outcome measures, risk perception), the patient's involvement in the decision-making process (e.g., self-efficacy and SDM process outcome measures), and the alignment of the final decision with the patient's values and preferences (e.g., decision conflict outcomes).

Thus, the primary objective of the review was to examine the evidence on the effectiveness of Shared Decision Making (SDM) tools specifically used for cancer screening. The review aimed to synthesize the evidence regarding how these tools impact the process of shared decision-making decision outcomes related to cancer screening. Please see attached manuscript revision, page 3, line 4

Point 6: Is there any difference between cancer types and outcomes?  i.e. was SDM better for breast and worse for prostate amongst populations?

Response 6:

  • Firstly, our subgroup analysis demonstrated a positive effect, indicating that the use of SDM tools improves knowledge outcomes in colorectal and prostate cancer screening. However, considerable heterogeneity remains evident in the results.
  • Secondly, SDM tools increased the intention to screen outcomes among people considering colorectal cancer screening, with a moderate level of heterogeneity observed.
  • However, no significant effects were observed on the intention to screen outcomes for other cancer types.
  • Finally, there is insufficient evidence for other outcomes between different cancer types.

For more information and detailed edits on the difference between cancer types and outcomes, please see:

  • attached supplementary file_3, Forest plots, Figure S2, S4, and S6,
  • revised manuscript, page 15, lines 345-347 for knowledge outcomes and page 18, lines 395-398 for intention to screen, by targeted cancer screening results.

Point 7: Would edit implications to state more heavily that it appears that SDM tools may have more benefit in vulnerable than non-vulnerable populations – there was no change in intention to screen among non-vulnerable people and in a world where all clinicians are strapped for time, would highlight that actually these tools may not be best for everyone. That feels really important to me.

Response 7: Thank you for providing this valuable feedback. Point 7's implications were revised to emphasize that SDM tools may have had more benefits in vulnerable populations than in non-vulnerable populations (Please see page 1 lines 16-24, lines 30-43 and pages 28-29, lines 756-778). We highlighted that these tools might not have been the best fit for everyone, particularly considering clinicians' time constraints. Your input was greatly appreciated, and we ensured to reflect these considerations in the revised version of the manuscript.
